# Pathohistological Findings after Bilateral Ovariectomy in Mares with Behavioral Problems

**DOI:** 10.3390/ani14192899

**Published:** 2024-10-08

**Authors:** Nadine Wolf, Joachim A. Hahn, Ingrid Walter, Yury Zablotski, Holm Zerbe, Tanja S. Witte

**Affiliations:** 1Equine Hospital Starnberg, 82319 Starnberg, Bavaria, Germany; na_wolf@gmx.at (N.W.); ja.hahn@web.de (J.A.H.); 2Institute of Morphology, University of Veterinary Medicine Vienna, 1210 Vienna, Austria; ingrid.walter@vetmeduni.ac.at; 3Equine Hospital, Ludwig Maximilian University Munich, 85764 Oberschleissheim, Bavaria, Germany; y.zablotski@med.vetmed.uni-muenchen.de; 4Clinic for Ruminants, Ludwig Maximilian University Munich, 85764 Oberschleissheim, Bavaria, Germany; h.zerbe@lmu.de

**Keywords:** equine ovary, behavior, immunohistochemical marker, anti-Müllerian hormone, granulosa cell tumor

## Abstract

**Simple Summary:**

Bilateral ovariectomy in mares with behavioral problems is a common long-term solution with high owner satisfaction. However, a pathohistological explanation for behavioral improvement after surgery is lacking. Therefore, bilaterally removed, clinically unremarkable ovaries from mares with behavioral problems were immunohistologically evaluated and compared with pathohistologically confirmed granulosa cell tumors. A complete data set including clinical history, clinical examination, serum anti-Müllerian hormone (AMH), and testosterone concentrations was analyzed as the basis for the pathohistological study. Immunohistochemical evaluation of Ki-67, AMH, aromatase, epidermal growth factor receptor, calretinin, and epithelial cadherin revealed no clear differentiation between large follicular structures of clinically unremarkable ovaries and cyst-like structures of neoplastic ovaries. Clinical data and success rate after bilateral ovariectomy of 85% were comparable with previous studies. Preoperatively measured serum AMH and testosterone concentrations were indicative of advanced granulosa cell tumors but were variable in mares with clinically unremarkable ovaries. Ultrasonographically nondetectable early neoplastic changes could be determined in 15% of mares and anovulatory-like follicles in 30% of mares with bilaterally removed ovaries. These changes might be a pathohistological explanation for behavioral problems of ovarian origin and a reason for the high success rate of bilateral ovariectomy.

**Abstract:**

Behavioral problems in reproductively healthy mares are a challenging issue that is successfully treated with bilateral ovariectomy (BO). This laparoscopic procedure represents an alternative to conservative treatment for mares not intended for breeding and results in high owner satisfaction regarding behavioral improvement. However, a pathohistological explanation to justify surgical ovarian removal regarding animal welfare is lacking. Therefore, the objective of this study was to pathohistologically evaluate bilaterally removed, clinically unremarkable ovaries of mares with behavioral problems (bOE, *n* = 20) and to compare them with pathohistologically confirmed granulosa cell tumors of mares with neoplastic ovaries (GCT-uOE, *n* = 10). A complete data set including preliminary presentation, clinical examination, and serum anti-Müllerian hormone (AMH) and testosterone was further analyzed in both groups. Both hormones were significantly higher in GCT-uOE compared with bOE. Immunohistochemical expression of Ki-67, AMH, aromatase, epidermal growth factor receptor, calretinin, and epithelial cadherin in granulosa cells of large follicular structures in bOE did not differ from neoplastic granulosa cells in GCT-uOE. Ultrasonographically nondetectable early neoplastic changes were pathohistologically evaluated in 15% of mares and anovulatory-like follicles in 30% of mares in bOE and might be one explanation for the high success rate of BO in 85% of bOE in this study.

## 1. Introduction

Mares with behavioral problems not obviously related to ovarian pathology are common and represent a challenge for owners, trainers, and veterinarians. As behavioral problems are assumed to be related to estrus, an association with ovarian hormonal secretion and anatomic changes is also assumed, especially in high-performance athletic mares. In some cases, even daily handling leads to difficulties and danger during estrus [1,2,3]. Consequently, veterinarians are expected to diagnose the cause of behavioral problems in order to find a successful solution for owners, riders, and the mares themselves. Behavioral problems range from normal, estrous-related sexual behavior due to physiological steroid hormone production, to management-related problems and medical issues caused by disease or pain of unknown origin [3,4,5]. Therefore, a multimodal approach with an assessment of all body systems with additional appraisal of management factors is recommended [6]. A possible medical cause for behavioral problems is the commonly occurring granulosa cell or granulosa-theca cell tumor, abbreviated as GCT in this study, due to excessive hormonal activity [7,8]. However, mares frequently present behavioral problems despite clinically unremarkable ovaries and a missing obvious ovarian neoplasia. Therefore, these cases pose a diagnostic and therapeutic challenge. 

Presuming an association of behavioral problems with the cyclicity of the mare in the absence of ovarian pathology, owners ask for methods to suppress ovarian function and estrous behavior. Aurich and Kaps (2022) [9], as well as Crabtree (2022) [10], currently summarize a variety of conservative treatment options for estrous suppression, but most of them are cost-intensive, might have severe side effects, or their reversibility is not clearly determined yet. Furthermore, legal limitations in Germany restrict the use of the majority of the conservative possibilities [11]. Therefore, bilateral ovariectomy (BO) is often inquired due to the minimally invasive surgical approach and safe method [12] for mares that are not intended for breeding. Although persistent estrous signs in mares after BO are described (in 35% [13], 27% [14], 20% [15], respectively), BO revealed a significant improvement of behavior after surgery from the owner’s perspective (success rate of 40% [5], 83% [16], 89% [17], 90% [14], 91% [18], respectively). However, an explanation for BO as a successful solution for mares with behavioral problems not obviously related to estrus or ovarian pathology is still lacking. 

Gonadectomy in male horses in order to facilitate daily handling, rideability, and cohabitation with other horses is common practice. However, the indication for BO in mares especially in case of a missing ovarian pathology or a lack of an understandable link to the ovaries [4] is discussed due to animal welfare aspects regarding removal of healthy organs [11]. Abdominal discomfort associated with physiological estrus is reported as “painful ovary syndrome” [3,10,19] and a justified reason for BO as a long-term solution. Further problems such as aggressive or stallion-like behavior are commonly associated with GCTs and treated by unilateral removal of the affected ovary [8,20]. However, these behavioral abnormalities were also found in mares with normal ovaries [17]. Therefore, the question arose as to whether bilaterally removed ovaries of clinically and reproductively unremarkable mares with behavioral problems might demonstrate other, non-neoplastic abnormalities that could explain the success and owner’s satisfaction after BO. 

Pathohistological examination using diagnostic markers is reported to effectively determine ovarian abnormalities, especially equine GCTs [21,22,23,24,25,26]. Dolin et al. (2023) [21] evaluated different immunohistochemical markers in a single equine GCT case, including Ki-67 (Ki67), anti-Müllerian hormone (AMH), aromatase (AR), calretinin (CAL), and epithelial cadherin (E-Cad), and suggested most of them as potential markers for tumor diagnosis. Ki-67 has tumor proliferation activity and is correlated to tumor progression, metastasis, and prognosis in humans [27,28,29]. Anti-Müllerian hormone is produced by granulosa cells (GCs) of postnatal females [30]. It is expressed in growing follicles of healthy equine ovaries and to a higher extent in GCTs [31,32,33]. Therefore, serum AMH is regarded as a sensitive marker for GCTs in mares [34,35]. The enzyme complex AR is expressed in healthy ovarian tissue and suggested as a further valuable diagnostic marker for equine GCTs [21,23,24,36,37]. Moreover, CAL, E-Cad, and epidermal growth factor receptor (EGFR) represent essential tumor markers, originally used in human tumor diagnostic [28,38,39]. Next to those tumor markers, CD56, GATA-4, and FOXL2 pose additional new diagnostic markers in human medicine as indicators for prognosis in GCTs [40]. Regarding the reported diagnostic value in humans, some of those immunohistochemical markers might also be helpful in detecting ovarian abnormalities other than GCTs in equine ovaries.

Therefore, the aim of the present study was to examine bilaterally removed ovaries of clinically unremarkable mares with behavioral problems (bOE) by means of histomorphology and immunohistochemistry. We hypothesized that GCTs are not the only ovarian source of behavioral problems in mares and that other, pathohistologically detectable changes might explain the reported success of BO. Moreover, we compared findings in bOE with findings in unilaterally removed, pathohistologically confirmed GCTs (GCT-uOE) and therefore hypothesized that clinically unremarkable ovaries of bOE are differentiated from neoplastic changes in GCT-uOE by means of the immunohistochemical markers Ki67, AMH, AR, EGFR, CAL, and E-Cad.

## 2. Materials and Methods

This pathohistological study was conducted in 2022 in accordance with national laws for animal use and approved by the ethics committee of the veterinary department of the LMU Munich (reference number: AZ 295-04-01-2022). The clinical part was performed from July 2019 to June 2022 and included clinical history, clinical examination, serum hormone analysis and surgical procedure. The clinical data set was retrospectively evaluated and implemented as basis for the pathohistological study. Surgical removal of the ovaries was suggested by referring veterinarians and performed with owner’s declaration of consent.

### 2.1. Animals, Group Determination, Clinical History, Premedication, and Preoperative Examination

Ovaries of 30 mares were collected after standing minimally invasive laparoscopic ovariectomy at the Equine Hospital in Starnberg. Mares were divided into two groups based on uni- or bilateral ovarian removal. In mares with unilateral ovariectomy (group GCT-uOE, *n* = 10 mares), GCT presence was suspected on clinical examination before surgery. In mares with BO (group bOE, *n* = 20 mares), both ovaries were unremarkable on clinical examination. This classification was further confirmed by routine pathohistological evaluation of removed ovaries by an external laboratory. Therefore, bilaterally removed ovaries (*n* = 40) of clinically unremarkable mares with behavioral problems (bOE) were compared with unilaterally removed, pathohistologically confirmed GCTs (GCT-uOE, *n* = 10). 

A detailed history was taken by means of an owner questionnaire prior to surgery (see Appendix A). The information included age, breed, duration of behavioral problems, behavioral patterns, and weather the mare had received any conservative treatment before surgery like Altrenogest (allyltrenbolone) or a GnRH vaccine (see Appendix A). A selection option of different behavioral patterns was provided according to commonly occurring behavioral abnormalities in mares (Appendix A) [5,17,18,41]. Further specifically mentioned behavioral problems observed by the owners themselves were also considered. All mares underwent a transrectal palpation and ultrasonographical examination of the reproductive tract before surgery to evaluate ovarian size, clinical presence of GCT, and stage of cycle determined by the presence of follicular structures and corpora lutea (CL). Estrus was defined if follicles > 30 mm were present in absence of a CL, diestrus if at least one CL was present [42], and an intermediate estrous stage if follicles > 30 mm and CL were ultrasonographically not detectable. The cyclic state was adapted if follicular structures or CL not identified on clinical examination were detected retrospectively by macroscopical or microscopical evaluation of the removed ovaries.

### 2.2. Hormone Concentrations

Routine blood collection was performed from a catheter immediately before surgery, centrifuged, and left over sera were frozen for further hormonal analysis. Serum AMH concentrations of all (*n* = 30) and testosterone concentrations of 18 cases (*n* = 11 in bOE, *n* = 7 in GCT-uOE) were determined by an external laboratory (Antech Lab Germany, formerly SYNLAB Vet) by means of a standardized and validated ELISA and LC-MS, respectively. In brief, AMH concentrations were measured using the Tecan Sunrise Absorbance Reader (AMH Gen II ELISA, Tecan, Männedorf, Switzerland), which works with a noncompetitive, two-side immunoassay. Analyses were performed according to the manufacturer’s instructions. Serum AMH concentrations between 14.3 pmol/L and 28.6 pmol/L were regarded as suspicious for GCT presence, while the cut-off value of 28.6 pmol/L was proving a GCT presence [35]. Testosterone concentrations were measured with the Sciex Triple Quad ^TM^ 5500+ System (Sciex, Framingham, MA, USA) using inductively coupled plasma mass spectrometry (ICP-MS) according to the manufacturer’s instructions. Serum testosterone concentrations above 0.35 nmol/L were indicating the presence of GCT. 

### 2.3. Surgical Procedure and Owner Consult

Standing laparoscopic ovariectomy was performed by the same surgeon using routine procedures [12]. In brief, mares were prepared for surgery 48 h in advance with no access to hay but to a special diet with haycobs, mash, and laxatives and were lunged several times a day. Mares received flunixin meglumine (1.1 mg/kg BW i.v., Flumeg Nova, Serumwerk Bernburg AG, Bernburg, Germany) before and until 5 days after surgery and were sedated (Detomidine 0.01 mg/kg BW i.v., Eurovet Animal Health B.V., Bladel, The Netherlands, and Butorphanol 0.01 mg/kg BW i.v., CP-Pharma, Burgdorf, Germany). After aseptic preparation, the incisional side on one (GCT-uOE) or both flanks (bOE) was infiltrated with local anesthetics (20 mL 2% lidocainhydrochloride, bela-pharm GmbH & Co. KG, Vechta, Germany). Three incisions for two instrumental portals and one optic portal were created on each side. The ovaries were detached at the anesthetized mesovarium using a vessel-sealing system (LigaSure^TM^, Medtronic, Meerbusch, Germany) and extracted through the extended flank incision, followed by suturing the incisions in the routine manner. All mares of bOE underwent a bilateral ovariectomy, whereas only the neoplastic ovary was removed in GCT-uOE. In the case of large-sized GCTs > 20 cm in diameter in GCT-uOE, a two-step procedure was performed with laparoscopic standing detachment followed by removal of the enlarged ovary in dorsal recumbency under general anesthesia [12]. Those mares were additionally treated with systemic antibiotics (Procain-Penicillin, 20.000 IE/kg BW i.m. SID, Dechra, Aulendorf, Germany, and Gentamicin, 6.6 mg/kg BW i.v. SID, CP-Pharma, Burgdorf, Germany) for 3 days. 

In both groups, a routine retrospective telephone survey was carried out 6 to 12 months after surgery regarding the improvement of behavioral problems, reoccurrence of behavioral problems, and possible signs of estrus, as well as owner’s satisfaction by means of a questionnaire (see Appendix A).

### 2.4. Macro- and Microscopical Evaluation of the Ovaries

Macroscopical examination of the removed ovaries including measurement of the size (length × width × height), gross section, and cycle stage determination was carried out immediately after the surgical procedure. Ovaries with 50–80 mm length and 20–40 mm width were regarded as normal sized [43], under 50 × 20 mm as small, and over 80 × 40 mm as large sized. Routinely, one representative sample of ovarian tissue with macroscopically visible follicles of each ovary was sent to a commercial pathohistological laboratory (Antech Lab Germany, formerly SYNLAB Vet Tierpathologie München, Munich, Germany) for routine pathohistological confirmation or exclusion of GCT. Additional samples with primarily visible follicles of the removed ovaries were fixed in 4% buffered formaldehyde and restored in 70% ethanol until further pathohistological analysis was conducted at the University of Veterinary Medicine Vienna (Vienna, Austria). There, specimens were embedded in paraffin, sectioned at 3–4 µm, and stained with hematoxylin and eosin (HE) for general tissue assessment. Slides of two different locations with primarily follicular structures were prepared for each ovary. Based on light microscopical examination, different stages of developing and regressing follicles of each ovary in bOE (*n* = 40) were categorized into preantral follicles (including primordial, primary, and secondary follicles), antral follicles (including tertiary and preovulatory follicles), atretic follicles (including early and late atretic follicles), and CL according to others [26,44,45,46]. Different GCT types of each ovary in GCT-uOE (*n* = 10) were routinely determined [25] and categorized according to the current World Organization for Animal Health (WOAH) tumor classification [47] but were not further included in the interpretation of results in this study. Macroscopical and pathohistological evaluation of removed ovaries was performed by the same trained observer advised by an experienced histologist and board-certified pathologists of an external laboratory. 

### 2.5. Immunohistochemical Staining and Evaluation

Consecutive serial sections of two different locations of each ovary were prepared for immunohistochemical examination with Ki67, AMH, AR, EGFR, CAL, and E-Cad. An indirect method with secondary antibodies conjugated with horseradish peroxidase was used. Therefore, paraffin sections of bOE and GCT-uOE were dewaxed through graded alcohol series (xylene, 100, 96, and 70% ethanol) and endogenous peroxidase activity was blocked by incubation in H_2_O_2_ with methanol at room temperature (RT) for 15 min with rinsing in tap water afterward. Table 1 summarizes functional and diagnostic relevance, sources, pretreatments, and dilutions of the primary antibodies, as well as equine positive controls used in this study. 

Antigen retrieval was processed by steaming in the presence of antigen unmasking solution (Tris-EDTA pH 9, or citrate buffer pH 6) for 30 min before cooling down to RT. Slides were rinsed in PBS and blocked with 1.5% normal goat serum (Sigma Aldrich, Merck, Darmstadt, Germany) for 30 min to minimize unspecific binding of the primary antibody. After blocking, sections were incubated with the primary antibody overnight at 4 °C, rinsed in PBS, and applied with BrightVision Poly-HRP-anti-rabbit second antibody (ImmunoLogic-Duiven-The-Netherlands) for 30 min at RT. After rinsing in PBS, slides were incubated with DAB-solution (Qanto, Richard Allan Scientific, TA-125-QHDX, Kalamazoo, MI, USA) according to the manufacturer’s protocol. Finally, nuclei were counterstained with haemalaun, rinsed in tap water, dehydrated in graded alcohol series (96 and 100% ethanol, xylene), and mounted with DPX medium (Fluka, Buchs, Switzerland) and cover glasses. Negative controls were obtained by omission of the primary antibodies to demonstrate the specificity of the secondary system. Sections of the equine lymph node, fetus, testes, skin, cerebrum, and kidney served as positive controls. Moreover, Western blots were performed to validate the specificity of the used antibodies for equine tissue (Appendix A). 

Immunolabeled slides were examined via light microscopy, and images were captured using a digital camera (UC90, Olympus, Munich, Germany) and imaging software (cellSense 2.3, Olympus). Immunoreactivity was assessed by evaluating four representative high-power fields per slide (200× magnification), and the number of positive cells of previously described different functional components of bOE and GCT-uOE was estimated. Expression of the proliferation marker Ki67 was evaluated by counting the number of positive stained nuclei in cells among a total of 100 cells, and the proliferation index (PI) was graded according to King et al. [27] as follows: 0 (PI 0–25%), 1 (PI 26–50%), 2 (PI 51–75%), and 3 (PI 76–100%). A high PI of Ki67 was set at >25% stained cells, including grade 1–3 [48]. Expression of AMH, AR, EGFR, CAL, and E-Cad was defined according to Ball et al. (2008) [31] with −, +, ++, and +++ for negative, mild, moderate, and high expression, respectively. 

For statistical evaluation of comparative immunohistochemistry, only GCs of detected large follicles with multiple GC layers (antral, early atretic, and anovulatory follicles) were included in bOE (*n* = 23) and compared with neoplastic GCs of GCT-uOE (*n* = 10), independent of the individual case. Expression of Ki67 was compared by means of the PI in percentage, whereas expression of the remaining markers was compared by means of their intensity. Moreover, immunohistochemical AMH expression was compared with serum AMH concentrations. Therefore, the highest detectable intensity of AMH in GCs of follicular structures in each case of bOE and in neoplastic GCs of cyst-like structures in each case of GCT-uOE was compared with the corresponding mare’s serum AMH concentration.

### 2.6. Statistical Analysis

Statistical analyses were performed using R version 4.3.3. (The R Foundation for Statistical Computing, Vienna, Austria). All parameters were tested for normality of distribution with the Shapiro–Wilk normality test. In cases where data were not normally distributed, the Mann–Whitney U test (Wilcoxon rank-sum test) for two groups and the Kruskal–Wallis test for more than two groups were conducted. In cases where the Kruskal–Wallis omnibus test was significant, additional Dunn pairwise tests were performed to demonstrate which groups differed significantly. 

For normally distributed parameters, Levene’s test was used to test the homogeneity of variances among groups. Student’s *t*-test for two groups and Fisher’s One-Way ANOVA for more than two groups were applied when variances were similar. Welch’s *t*-test and Welch’s ANOVA were used when variances differed. Subsequently, a pairwise Student’s *t*-test was additionally performed with Fischer’s ANOVA, and pairwise Games–Howell tests were used for Welch’s ANOVA to demonstrate which groups differed significantly. 

A chi-square test was performed to compare categorical parameters. In cases where one of the categorical variables had more than two categories, pairwise Fisher tests were performed. Statistical significance was set at *p* < 0.05. In the case of multiple comparisons (e.g., ANOVA, Kruskal–Wallis, pairwise Fisher tests), the *p*-values were adjusted with the Holm correction method for multiple testing.

## 3. Results

Findings of clinical history (behavioral patterns, duration of behavioral problems, and conservative treatment before surgery), clinical examination (serum AMH and testosterone concentration, rectal palpation, and ultrasonographical examination with determination of cyclic stage), outcome of surgery, and pathohistological evaluation of each case in bOE and GCT-uOE are summarized in Appendix A. Furthermore, parts of the study were previously published on the ISER (International Symposium on Equine Reproduction 2023) conference as a poster and published as an abstract in the Journal of Equine Veterinary Science [49].

### 3.1. Animals, Clinical History, and Pre-OP Examinations

Included mares were of different breeds, mainly German Warmblood (*n* = 16) and Polish Warmblood (*n* = 3), in addition to Arab, Haflinger, Frisian, Quarter Horse, Pure Raza Española, Trotter, Icelandic Pony, and other Ponies. In bOE, mares were 4–20 years old at presentation (mean 13.0 years, SD 3.7), whereas mares in GCT-uOE were aged from 8–16 years (mean 13.2 years, SD 5.0). 

Nineteen of twenty mares in bOE (95%) presented one or more specific behavioral problems (Appendix A). One mare in bOE was bilaterally ovariectomized due to an increased serum AMH concentration without reported behavioral problems (Case 16). Nine of 10 mares in GCT-uOE demonstrated at least one specific behavior pattern (Appendix A). One mare in GCT-uOE was ovariectomized due to a randomly detected neoplastic ovary on rectal palpation with normal cyclicity and behavior (Case 12). Table 2 provides an overview of the frequency of behavioral problems in bOE and GCT-uOE from the owner’s perspective. Moodiness and stressed manner were additionally mentioned as further behavioral disorders by owners and included in the classification. Moodiness was the most frequently complained about behavioral problem in mares of bOE, whereas stallion-like behavior was most commonly present in mares of GCT-uOE. As demonstrated in Table 2, behavior patterns differed among bOE and GCT-uOE. Aggressive behavior and increased flank sensitivity were not reported in any case of GCT-uOE, in contrast to bOE, with an incidence of 35% and 25%, respectively.

The time from the first presentation of behavioral problems to referral was under 12 months in 63% and over 12 months in 37% in bOE. In GCT-uOE, 67% of the mares showed behavioral problems under 12 months and 33% over 12 months before surgery. One case of each group was not incorporated due to no behavioral problems present before surgery (Case 16, 12). Eleven mares in bOE (55%) were conservatively treated either with Altrenogest (*n* = 5), GnRH-vaccine (*n* = 3), or Altrenogest following GnRH vaccination (*n* = 3) before surgery. All but one of the mares (91%) responded well to conservative treatment. In GCT-uOE, no mare was treated with Altrenogest or GnRH vaccination before surgery.

In bOE, rectal palpation prior to and macroscopical evaluation after surgery revealed symmetric ovaries in 16 mares (80%) with normal size (65%) and small size (15%) but asymmetric ovaries in 4 mares (20%). Four mares were assigned to estrus (20%), ten mares to diestrus (50%), and six mares to the intermediate estrous stage (30%). In GCT-uOE, all neoplastic ovaries were unilaterally enlarged on rectal palpation and macroscopical evaluation. All GCT-affected ovaries presented a honeycomb-like appearance on ultrasound and a small, inactive contralateral ovary. One mare presented a second GCT four years after the surgical removal of the first one (Case 37). 

### 3.2. Results of Hormone Measurements

Prior to surgery, serum AMH concentrations in bOE ranged from 0.4–43.5 pmol/L with a median of 11.8 pmol/L (interquartile range (IQR) 9.5 pmol/L). Seventy percent of serum AMH concentrations were within normal limits in bOE, whereas fifteen percent were in the suspicious range of 14.3–28.6 pmol/L and fifteen percent above the cut-off limit of 28.6 pmol/L. Serum AMH concentrations in GCT-uOE ranged from 123.0–150.0 pmol/L with a median of 143.0 pmol/L (IQR 19.3 pmol/L). Mares of GCT-uOE showed significantly higher serum AMH concentrations compared with bOE (*p* < 0.001), which is demonstrated in Figure 1.

Serum testosterone of bOE was measured in 11/20 cases (55%) before surgery and was within normal limits (<0.14 nmol/L) in 10 of 11 mares (95%). The only mare with increased serum testosterone of 0.45 nmol/L in bOE additionally presented increased serum AMH (38.6 pmol/L, Case 29). In GCT-uOE, serum testosterone concentrations were measured in 7/10 cases (70%) and were increased in 20% of those cases with a median of 0.21 nmol/L (range 0.14–1.25 nmol/L, IQR 0.42). Serum testosterone concentrations were significantly higher in GCT-uOE compared with bOE (*p* = 0.04). 

### 3.3. Outcome of Surgery and Improvement of Behavior

Surgeries in both groups were conducted throughout the year. Standing laparoscopic ovariectomy was performed in all mares (*n* = 30). Additional laparotomy under general anesthesia was necessary in five cases of GCT-uOE (50%) due to the large size of the neoplastic ovary. Overall complications were uncommon and present in 16.7% of all mares with fevers up to 39.7 °C in the first two days after surgery (16.7%) and seroma formation (6.7%) in both groups. Mild-to-moderate subcutaneous emphysema around the incision was found in all cases. Mares with standing surgery in bOE were discharged from the hospital after 5 to 7 days, whereas mares with enlarged GCTs in GCT-uOE following a two-step procedure or mares with complications in both groups were discharged after 7 to 10 days in a healthy condition. 

Routine owner follow-up 6 to 12 months after surgery resulted in an improvement in behavioral problems in 85% of bOE and in 100% of GCT-uOE (overall improvement in both groups 90%). Two mares of bOE (Case 39, 49) demonstrated colic signs before surgery and were euthanized due to recurrence of colic 1 and 6 months after surgery, respectively. One mare of bOE (Case 40) showed a reoccurrence of initial behavioral problems 6 months after surgery with aggressiveness towards other horses, unwillingness to be ridden, and additional persistent estrous signs. Persistent estrous signs after BO occurred in 10% of the mares and included Case 40 of bOE and Case 37 of GCT-uOE. Case 37 was ovariectomized due to a second GCT. Both owners described the residual estrous signs as mild and well manageable. Of all conservatively treated mares before surgery in bOE, 4/5 (80%) were treated with Altrenogest, 2/3 (67%) were treated with GnRH-vaccination, and 2/3 (67%) were treated with both further improved after BO.

### 3.4. Macro- and Microscopical Findings on the Ovaries

Size determination of extracted ovaries in bOE revealed the largest ovary with 80 × 40 × 30 mm and the smallest ovary with 35 × 20 × 15 mm. Normal-sized symmetric ovaries were determined in 65%, small-sized symmetric ovaries in 15%, and asymmetric ovaries of normal and small size in 20% in bOE. In gross sections, numerous multiple follicles, antral follicles, and CL (Figure 2(A1–A3)) were found. Moreover, a small pale area near the ovulation fossa in one ovary in bOE (Case 29, Figure 3A, red circle) was detected. All GCT-affected ovaries in GCT-uOE were large-sized with a small-sized contralateral ovary. The largest GCT was not measurable due to intraoperative fragmentation but weighed approximately 5 kg (Case 28, Figure 2(B1)). The smallest ovary of GCT-uOE measured 85 × 65 × 55 mm. Gross sections revealed a multicystic appearance in all neoplastic ovaries of GCT-uOE (Figure 2(B1–B3)). 

In bOE, microscopical evaluation of removed ovaries (*n* = 40) revealed preantral follicles (primordial, primary, and secondary follicles) in 40%, antral follicles (tertiary and preovulatory follicles; Figure 4(A1–B7)) in 30%, early atretic follicles in 28%, late atretic follicles in 40%, and CL in 28%. We could additionally determine large follicular structures with the size of preovulatory follicles and multiple GC layers but a poorly developed theca cell layer (Figure 5D). This layer was mainly formed by theca cells with a retained fibroblast-type appearance in contrast to preovulatory follicles, which contained polyhedral cells in a well-developed and vascularized theca interna cell layer (Figure 5A). These large follicles were defined as anovulatory-like follicles according to the histomorphological definition of anovulatory follicles [50] but without clinical examination of persistence. These structures were present in 15% of all examined ovaries and in 30% of all mares of bOE. All follicles with multiple GC layers were summarized as large follicles and included antral follicles, early atretic follicles, and anovulatory-like follicles. Polyhedral cells with a pale nucleus and foamy cytoplasm could occasionally be seen in the theca interna of preovulatory follicles (Figure 4(B1–B7), asterisks) and early atretic follicles. Fossa cysts with flattened, partially ciliated epithelial cells were present in 25% of all examined ovaries and in total in 35% of all mares in bOE. Small areas of GC nests (Figure 3(B1,B2)) or spindle-shaped GCs in the GC layer (Figure 3(C1–C6)) were detected in four ovaries (10%) of three mares in bOE (15%). Moreover, polyhedral cells with foamy cytoplasm (similar to Leydig-like cells in GCTs) could be found in the theca interna cell layer (Figure 3(C1–C6)) or within GC nests (Figure 3(B1,B2), asterisks) in those cases. Such histological abnormalities were defined as early neoplastic changes (ENCs) in this study and regarded as the onset of ovarian degeneration. Early neoplastic changes occurred unilaterally (Case 10, 36), as well as bilaterally (Case 29), and were growing from large follicular structures and mainly from anovulatory-like follicles (75% of ENC). 

All tissue samples of GCT-uOE demonstrated a cyst-like appearance with variable structural compositions of macrofollicular, microfollicular, insular, trabecular, and diffuse types of patterns. Slim, prismatic GCs with Sertoli-cell morphology, so-called Sertoli-like cells (SLCs), and numerous polyhedral LLCs with foamy cytoplasm in the theca cell layer were detected in 70% of GCT-uOE. Leydig-like cells were invasive growing by penetrating the basement membrane into the GC layer in one ovary of GCT-uOE (Case 24, Figure 5B, asterisk). 

### 3.5. Immunohistochemical Evaluation

The PI grade of Ki67 and intensity of immunohistochemical expression of AMH, AR, EGFR, and CAL in different cell populations of ovarian structures in bOE and GCT-uOE are summarized in Appendix A. Evaluated cell populations included GCs, theca cells, and lutein cells in bOE and GC, LLC, SLC, and theca cells in GCT-uOE. In brief, all tested markers, with the exception of E-Cad, were expressed in the ovaries of bOE and GCT-uOE. The proliferation marker Ki67 was mainly expressed in nuclei, whereas AMH, AR, EGFR, and CAL were generally allocated to nuclei and/or cytoplasm. Epithelial cadherin showed delicate cell membrane-associated staining and was restricted to epithelial cells of fossa cysts and the zona pellucida of oocytes (Figure 4(A6), arrow). Epithelial cadherin was not expressed in other structures of bOE or neoplastic tissue of GCT-uOE and was consequently not further evaluated. 

To differentiate between clinically unremarkable ovaries of bOE and neoplastic ovaries of GCT-uOE, the GCs of large follicular structures of bOE, including antral follicles, early atretic follicles, and anovulatory-like follicles, were statistically compared with neoplastic GCs of cyst-like structures of GCT-uOE. Therefore, only evaluation of those cells was included for further discussion. A complete immunohistochemical analysis of all detected structures in bOE and GCT-uOE is provided in Appendix A. 

#### 3.5.1. Expression of Ki67, AMH, AR, EGFR, and CAL in Large Follicular Structures of bOE

Granulosa cells of tertiary follicles (Figure 4(A1–A6)) showed a varying Ki67 PI (grade 1–3), moderate AR and CAL expression, and intense expression of AMH and EGFR. Granulosa cells of preovulatory follicles (Figure 4(B1–B7)) presented a high Ki67 PI (grade 2–3), moderate CAL, and intense expression of AMH, AR, and EGFR. Polyhedral cells of the well-developed theca interna layer of preovulatory follicles (Figure 4(B1–B7), asterisks) revealed high AR and CAL and mild AMH and EGFR expression. Granulosa cells of early atretic follicles showed a high Ki67 PI (grade 2–3) and high expression of AMH, AR, EGFR, and CAL, similar to preovulatory follicles. Granulosa cells of anovulatory-like follicles (Figure 5D) presented a varying Ki67 PI, and AMH, AR, EGFR, and CAL were less expressed compared with GC of preovulatory follicles. Early neoplastic changes showed a low Ki67 PI, mild AMH, moderate CAL, high AR, and high EGFR expression in spindle-shaped GCs, as well as in LLCs in the theca interna cell layer (Figure 3(C1–C6)) or in between GC nests (Figure 3(B1,B2)), with no difference to adherent healthy GC tissue. 

#### 3.5.2. Expression of Ki67, AMH, AR, EGFR, and CAL in Neoplastic Cells of GCT-uOE

Neoplastic GCs of GCT-uOE were characterized by high expression of EGFR and, in general, moderate AR, AMH, and CAL coexpression with partly heterogeneous patterns within and between the tumors. The proliferation marker Ki67 was varyingly expressed in neoplastic GCs with a high PI in 30% and a low PI in the remaining 70% of GCT-uOE (Figure 5B,C, Ki67). Leydig-like cells (Figure 5B) demonstrated high coexpression of AR, EGFR, and CAL and moderate AMH expression. Sertoli-like cells presented an expression pattern equal to neoplastic GC. 

#### 3.5.3. Comparative Immunohistochemistry between bOE and GCT-uOE

Comparative immunohistochemistry between GCs of large follicular structures in bOE (*n* = 23, Figure 5A,D) and neoplastic GCs of cyst-like structures in GCT-uOE (*n* = 10, Figure 5B,C) revealed a statistically significant higher Ki67 PI (*p* = 0.02) in bOE (median 51.0%, IQR 52.0) compared with GCT-uOE (median 4.5%, IQR 27.3%). Immunohistochemical analysis of AMH, AR, EGFR, and CAL revealed no significant difference between the two groups (*p* > 0.05).

#### 3.5.4. Correlation of Serum AMH and Immunohistochemical AMH Expression 

In bOE, a significant correlation between serum AMH concentrations and AMH expression of follicular structures detected by immunohistochemistry was evaluated (*p* = 0.02). No correlation was found between serum AMH concentrations and immunohistochemical AMH expression in neoplastic GCs of GCT-uOE (*p* = 0.25).

## 4. Discussion

Bilateral ovariectomy in mares with clinically unremarkable ovaries but behavioral problems is known to result in high improvement of behavior from the owner’s perspective [5,15,16,17,18]. However, an underlying pathohistological reason for this phenomenon has not been identified yet. To our knowledge, this is the first pathohistological study characterizing bilaterally removed, clinically unremarkable ovaries of mares with behavioral problems in comparison with pathohistologically confirmed GCTs by means of histomorphology and immunohistochemistry. We further involved a complete clinical data set to confirm the positive outcome of bilateral ovariectomy in 85% of mares in our study, which is in accordance with recent studies [5,15,16,17,18]. We, therefore, included a detailed clinical history, clinical examination, and analysis of serum AMH and testosterone concentrations as the basis for the pathohistological study. Evaluation of the complete clinical data resulted in a wide variety of each examined mare in bOE and GCT-uOE and is therefore summarized in Appendix A.

Conservative treatment with Altrenogest and/or GnRH vaccination before surgery resulted in a good response in 91% (10/11) of treated mares of bOE. Therefore, ovaries are suggested as the source of behavioral problems in these mares. Commonly, both treatment options are used in mares with estrous-related behavioral problems but are restricted in Germany as a long-term solution [11]. Moreover, Altrenogest was reported to be less effective compared with BO regarding behavioral improvement, but a correlation with a positive outcome of BO could be seen [15]. In contrast, Collar et al. (2012) [14] found no efficient prediction for the behavioral outcome of BO by means of Altrenogest administration before surgery. In our study, 73% (8/11) of the conservatively treated mares showed a positive outcome of BO, but this included mares that were treated with Altrenogest, GnRH vaccination, or both, and the exact date and duration of administration remained unknown. GnRH vaccination has a reported significant impact on ovarian activity and size [51,52] and was even reported to decrease serum testosterone and ovarian size in three GCT-affected mares with additional resolution of behavioral problems [53]. Therefore, GnRH vaccination in six mares of bOE in our study might have suppressed presumptive present abnormalities, as histomorphological evaluation revealed no pathological findings in those mares.

The time from onset of behavioral problems to referral was less than 12 months in 63% of bOE and in 67% of GCT-uOE and was therefore comparable to outcomes of Straticò et al. (2023) [18], who found no difference in time of presentation and severity of behavioral problems of mares with normal ovaries and mares with GCTs. These results, moreover, emphasize that behavioral problems occurred recently and suggest the development of a pathological event in those mares. In the case of long-term behavioral problems in 37% of bOE, the possibility of learned behavior or even a pre-existing problem [4], especially in older mares, should be considered as well. However, we also determined long-lasting behavioral problems in young mares (Case 10, 4 years old) with detected ENCs as pathological events, which was uncommon in our results.

As already detected by others [17,18], the mares in our study presented in general more than one specific behavioral problem, with moodiness and unwillingness to be ridden most commonly occurring in bOE and stallion-like behavior predominantly present in GCT-uOE (Table 2). Stallion-like behavior is characterized by attempts to mount other mares [4,8] and was observed in one case of bOE with pathohistologically determined ENCs (Case 10) and in 60% of GCT-uOE. This pattern is regarded as a testosterone-driven-sexual behavior and, together with an aggressive manner, is reported as pathognomonic for GCTs [8,54]. However, serum testosterone concentrations were evaluated only in single mares and therefore associated with stallion-like behavior in only one mare of GCT-uOE (Case 32). Aggressive behavior was clearly separated between aggression towards people and towards other horses in our study (Table 2, Appendix A), but neither was observed in any case of GCT-uOE, including cases with increased serum testosterone concentrations. These findings therefore support the outcome of Huggins et al. (2023) [41], who could not find a significant trend between aggressive behavior and increased serum hormone concentrations.

Serum AMH concentrations of clinically unremarkable mares in bOE varied from 0.4 to 43.5 pmol/L. With a median of 11.8 pmol/L in bOE, serum AMH concentrations in our study were higher compared with reported concentrations of 6.9 pmol/L (mean) in cyclic mares [34] and 2.14 pmol/L (median) in reproductively normal mares [35]. Serum AMH is a highly sensitive marker for GCTs and independent of daily fluctuations, cyclic stage, or pregnancy [34,35,55,56,57]. However, the inclusion of mares in all cyclic stages and with constant or prolonged estrus (15%) or anovulatory-like follicles (30%) and ultrasonographically nondetectable, small nests of neoplastic GCs (ENCs, 15%) in one or both ovaries of bOE might explain the deviating median serum AMH in our study compared with others [34,35]. Early neoplastic changes might contribute to variable serum AMH concentrations in bOE, as two mares with pathohistologically detected ENCs presented increased serum AMH concentrations (Case 29, 36), whereas the third mare (Case 10) showed the lowest measured AMH concentration of 0.4 pmol/L in bOE. Wide fluctuations of periodically measured serum AMH within specific cases have been reported [58], and serum AMH might not be reliable in the case of early tumor growth [17]. Devick et al. (2020) [17] reported an elevation of AMH above 27.0 pmol/L in only 44% and above 57.0 pmol/L in only 11% of pathohistologically confirmed GCTs and concluded there was a low sensitivity of AMH in early detected GCTs, which is in accordance with our findings. Therefore, repeated measurement of serum AMH is recommended in ambiguous cases [59]. Furthermore, Devick et al. (2020) reported perceived false positive serum AMH measurements, as investigated mares with pathohistologically normal ovaries had increased concentrations above 27.0 pmol/L in 53% and above 57.0 pmol/L in 24% in their study. Comparably, 15% of bOE in our study were in the suspicious range above 14.3 pmol/L and 15% above the cut-off value of 28.6 pmol/L. However, the suspicious range and the cut-off value used by our laboratory were lower compared with others [17,41]. Ball et al. (2013) [35] determined concentrations of over 28.6 pmol/L, indicating GCT presence, which was in accordance with our cut-off value, but the authors did not include a suspicious range. Five of the six mares with increased serum AMH in bOE showed behavioral abnormalities with pathohistologically detected ENCs (*n* = 2), anovulatory-like follicles (*n* = 4), and fossa cysts (*n* = 1). The remaining mare with increased serum AMH (Case 16), however, showed no behavioral problems and no abnormalities on pathohistological evaluation. Regarding the cut-off value of 28.6 pmol/L [35], an increased AMH of 43.5 pmol/L in Case 16 would therefore prove GCT presence. Consequently, based on our results, the reference limits of 14.3 pmol/L and 28.6 pmol/L used in this study were adjusted by the laboratory, with concentrations above 71.4 pmol/L clearly indicating the existence of GCTs and less than 30.7 pmol/L suggesting healthy ovaries. These new reference values are comparable to values used by Huggins et al. (2023) [41]. As all mares in GCT-uOE showed significantly higher serum AMH concentrations compared with bOE, with a range of 127.0–150.0 pmol/L (Figure 1), we assume neoplastic GCs as the source of increased serum AMH in GCT-uOE, in accordance with others [31,34]. However, a statistical correlation between high serum AMH concentrations and immunohistochemical AMH expression could not be found in GCT-uOE. Increased serum AMH in mares with GCTs might therefore be caused by the abnormally high number of proliferative cells forming tumors of advanced size, similar to inhibin [22]. This fact could further explain the low serum AMH despite the pathohistological presence of ENCs in Case 10 of bOE, as the total number of neoplastic GCs was still low. 

Serum testosterone measurements also resulted in significantly higher concentrations in GCT-uOE compared with bOE, although partly tested (bOE 55%, GCT-uOE 70%). The only case in bOE with increased serum testosterone revealed additional high serum AMH (Case 29), which could be explained by pathohistologically detected ENCs with LLCs present on both ovaries (Figure 3(B1,B2)). Elevation of serum testosterone is commonly associated with stallion-like behavior and/or the presence of a substantial theca cell component and LLCs in GCTs [8,22,24,25,56,60,61]. This was observed in all mares with increased testosterone concentrations of GCT-uOE (*n* = 2), as well as in Case 29 of bOE. Although the number of LLCs between neoplastic GC nests was still low (Figure 3(B1,B2)), we assume an association between the presence of LLCs and elevated serum testosterone in Case 29. Increased serum AMH and testosterone concentrations were suspicious for ovarian dysfunction or GCT existence in Case 29, but clinical examination revealed reproductive functionality in this mare due to CL presence on one ovary. This case therefore represents a diagnostic challenge, as serum AMH and testosterone concentrations might be unpredictable or even vary over time [58]. A transvaginal ovarian biopsy sample [62] could be helpful in determining which ovary is affected in such unclear cases or in mares where breeding is still desired. However, this invasive method might result in false negative outcomes in the case of widely distributed ENCs. Bilateral ovariectomy was successful in Case 29 regarding behavioral improvement and, moreover, efficient in the removal of pathological ovaries with bilaterally occurring ENCs, which were clinically unremarkable. Conclusively, routine diagnosis by means of clinical examination and serum hormone analysis might be unclear in some cases [58], especially in early tumor growth [17].

Due to the high success rate of bOE in our study (85%) and others [5,14,15,16,17,18], ovaries are suspected as the main cause of behavioral problems, although a detailed diagnostic workup to rule out other, abnormal behavior-causing issues [4,6] was mostly lacking in our study and others [5,14,15,16,17,18]. Therefore, other clinical causes for behavioral problems remained unknown before surgery, and a direct relation to estrus could not be determined by clinical history in our study. However, the positive response to preoperative conservative treatment was indicative of an association with estrus, which is moreover supported by the highly successful outcome of BO. Accordingly, Melgaard et al. (2020) [5] and Kamm and Hendrickson (2007) [16] found an improvement in behavior after BO not obviously related to the estrous cycle. Comparable to our results, Collar et al. (2021) reported a resolution of the initially presenting problems in 90% of 41 elective cases, whereas Melgaard et al. (2020) revealed an improvement of behavior of 40% and in rideability of 80% in 10 mares with pathohistologically normal ovaries. Furthermore, others reported similar high success rates of BO in mares with behavioral problems (83% [16], 89% [17]) but also included cases with other medical problems or pathohistologically diagnosed GCTs. Our overall success rate of ovariectomy in bOE and GCT-uOE was in a range of 85–95%, which is even higher but difficult to compare with others [16,17], as mares with pathohistologically diagnosed GCTs were only unilaterally ovariectomized in our study. All mares of GCT-uOE showed normal behavior after removal of the affected ovary, resulting in complete owner satisfaction and confirming the proven common cause for behavioral problems by endocrinologically active GCTs [8,20,60]. Owners were not satisfied with BO in only 15% (3/20) of bOE. This included two cases with recurrent colic signs (Case 39, 49), which might have been misattributed to the ovaries [4] and represented other, not-ovarian-related reasons for behavioral problems in those mares [6]. The third mare (Case 40) showed recurring behavioral problems simultaneously with persistent estrous signs. Persistent estrous signs after BO are described in 20–35% of cases [13,14,15]. In our study, the remaining estrous signs were with an incidence of 10% low and reported as mild by the owners. However, owners should be aware that BO might not be successful in all mares, especially if abnormal clinical findings and increased hormonal concentrations are absent [58]. Complications after BO were limited to 16.7% of all cases in our study with fevers (up to 39.7 °C) and seroma formation, as well as mild-to-moderate emphysema, suggesting this surgical procedure as a safe method [12]. 

Macroscopical evaluation of the removed ovaries in bOE was in accordance with rectal palpation and ultrasonographical examination regarding structure and size. However, Case 29 of bOE presented a suspicious pale area near the ovulation fossa on gross examination (Figure 3A, red circle), which was nondetectable on ultrasonographical examination. This area revealed nest-forming GCs with polyhedral, foamy cells resembling LLCs (Figure 3(B1,B2)) on pathohistological examination and were determined as ENCs. Such ENCs were histomorphologically also present as spindle-shaped GCs with LLCs in the theca interna cell layer of bOE (Case 10, Figure 3(C1–C6)). Early neoplastic changes are reported as early-stage GCTs by others [17], but a detailed histomorphological description is lacking. In bOE, ENCs were detected in functional ovaries with additional CL on the same (Case 29) or the contralateral ovary (Case 10) or with large follicular structures present (Case 36). Mares with developing GCTs are reported to continue normal estrous cyclicity or even to maintain pregnancy with the presence of a functional CL [63,64,65,66]. Consequently, a transitional period between the clinical manifestation of neoplastic GCs initially and loss of normal estrous cycle is assumed and explained by a lack of inhibitory effect of inhibin on the contralateral ovary [35,67]. Mares in this transitional period might already present abnormal behavior with no abnormalities on clinical examination and variable serum AMH and testosterone concentrations (Case 10, 29). Therefore, clinically nondetectable ENCs could be one explanation for the high success rate of BO by causing behavioral abnormalities with no clear detection by routine diagnosis. Anovulatory-like follicles were determined in 30% of the mares in bOE according to their histomorphological similarity to anovulatory follicles (Figure 5D, [50]). Anovulatory follicles are classified as persistent anovulatory follicles with limited clinical significance or luteinized hemorrhagic anovulatory follicles with an assumed responsibility for infertility in mares [68,69,70,71]. Clinical evaluation of persistence of anovulatory-like follicles was lacking in our study, but their histomorphological presence in 30% of the mares in bOE was regarded as pathological due to their high incidence compared with a reported 8.2% in clinically confirmed, persistent anovulatory follicles [68]. As 75% of ENCs were growing from anovulatory-like follicles, we further suggest these structures as possible precursor stages for ENCs and advanced GCTs [58,65]. Two mares with detected anovulatory-like follicles presented colic signs (33%), which were therefore considered “painful ovary syndrome” [3,10,19], whereas others showed mixed patterns of behavioral problems with the conclusion of no direct association to a special behavioral problem. However, anovulatory-like follicles might be a further explanation for successful BO, as 87% of the mares in bOE with those defined structures showed behavioral improvement after BO (Appendix A). Histomorphological analysis revealed no abnormalities in 65% of the mares in bOE (13/20). Fossa cysts were frequently detected in bOE (35%) but not regarded as pathological due to their common occurrence in equine ovaries [47,70]. Those nonfollicular cysts are unlikely to cause behavioral problems but might trigger the development of anovulatory hemorrhagic follicles [70,71].

The typical macroscopical and histomorphological presence of GCTs in all mares of GCT-uOE with large size, ultrasonographically multicystic, honeycomb-like appearance (Figure 2(B1–B3)), and histological patterns were comparable to the results of others [25,54,60]. A similar presence of LLCs and SLCs in GCT-uOE at 70% was also found in other studies [22,25,54,56,60]. Leydig-like cells were located between neoplastic GCs or in the theca cell layer (Figure 5B), and some of them presented an invasive growing character (Case 24, Figure 5B, asterisks), which was not described by others before. Early neoplastic changes in bOE presented LLCs either between GC nests (Figure 3(B1,B2)) or in the theca cell layer (Figure 3(C1–C6)) and were hardly distinguishable from polyhedral cells in the theca interna of preovulatory follicles (Figure 4(B1–B7), asterisks) by means of immunohistochemical analysis (Appendix A). In general, ENCs revealed similar immunohistochemical patterns to adjacent healthy ovarian tissue of bOE and neoplastic GCs of GCT-uOE and could only be determined by their histomorphological appearance. 

Immunohistochemical comparison between ovaries of bOE and GCT-uOE revealed no evident difference in the expression of Ki67, AMH, AR, EGFR, CAL, and E-Cad. Immunohistochemical expression patterns of GCs of large follicular structures in bOE resembled those of neoplastic, cyst-like structures of GCT-uOE (Figure 5). This was in accordance with others, where further markers like inhibin were tested [22,25,45]. In contrast, Dolin et al. (2023) [21] reported expression patterns of AMH, AR, CAL, and E-Cad in neoplastic tissue that differed from normal ovarian tissue but were evaluated in only a single GCT. We evaluated 40 clinically unremarkable ovaries and compared them with 10 pathohistologically confirmed GCTs but revealed neither an immunohistologically visible (Figure 5) nor statistical difference between the two groups, with the exception of Ki67. The significantly higher Ki67 PI in large follicles of bOE (median 51.0%) compared with cyst-like structures in GCT-uOE (median 4.5%) emphasizes the high proliferation activity in these large, non-neoplastic follicular structures. Granulosa cell tumors are regarded as predominantly benign and well differentiated [7,22,25]. Therefore, the mild expression of Ki67 in GCT-uOE (Figure 5B,C, Ki67) indicates a low proliferation rate [21,48]. Immunolabeling of AMH in both groups was similar to other reports [21,31]. The high AMH expression in GCs of preantral and early atretic follicles in bOE (Appendix A) could explain the positive correlation (*p* = 0.02) between AMH expression detected by immunohistochemistry and serum AMH in bOE. Aromatase was highly expressed in LLCs (Figure 5B, AR) but showed a heterogeneous pattern in neoplastic GCs of GCT-uOE (Figure 5B,C, AR), according to Dolin et al. (2023) [21]. A previously reported high CAL expression in neoplastic GCs and LLCs [21] was also present in GCT-uOE, with additional high expression of EGFR, which was not evaluated before. The adhesion molecule E-Cad was restrictively expressed in the zona pellucida of oocytes (Figure 4A6, arrow) and in epithelial cells of fossa cysts of bOE and therefore indicative of those structures. However, we doubt the reported diagnostic value of E-Cad for GCT detection [21], as this marker was not present in any GCTs of GCT-uOE. According to recent studies regarding the prognosis for human GCTs, immunohistochemical markers like CD56, GATA-4, and FOXL2 were associated with reduced prognosis [40]. These markers might also be useful in veterinary medicine but have to be validated first. 

The main limitations of this study include a low caseload of 20 mares and a total of 40 examined ovaries in bOE, but these numbers are comparable to similar studies [15,16,17,18]. A control group of healthy mares without behavioral problems and normal serum hormone concentrations, as well as physiological ovaries, was not included. The main objective of our study was to compare clinically unremarkable ovaries of mares with behavioral problems with GCTs. Furthermore, we concentrated on serum AMH and testosterone concentrations, as these hormones, together with inhibin, are reported as main indicators for behavioral changes [41]. However, testosterone concentrations were evaluated only in some mares. A complete hormonal panel, including progesterone and estradiol with repeatable measurements throughout the cycle, would help to gain more information regarding the mare’s reproductive health. In the pathohistological evaluation, large follicular structures were determined in 58% of the examined ovaries and resulted in a smaller caseload to compare immunohistochemical expression patterns of bOE with GCT-uOE. As the clinical study was conducted independently of the pathohistological study, 24% of the ovaries (*n* = 8 in bOE, *n* = 3 in GCT-uOE) were buffered in formaldehyde up to 18 months before histological preparation was started and therefore resulted in a lack of detectable Ki67 expression. The assessment of behavioral problems by means of detailed clinical history and evaluation of behavioral improvement after 6 to 12 months by a retrospective telephone survey are commonly used for evaluating the owner’s opinion [5,14,16,17,18] but might reflect only a subjective opinion of the owner. This could be reinforced by the fact that scoring of behavioral problems before and after surgery [5,18] was missing in our study. The lack of information on GnRH vaccination timepoints further complicates the interpretation of clinical examination and pathohistological findings in mares conservatively treated before surgery. Moreover, information regarding the duration of ownership or assessment of other body systems [6] before referral was not available. Although not the aim of this study, this additional information could be helpful in determining non-ovarian-related issues and explaining why BO might fail in some mares (15% of bOE). 

Comprehensive studies with a focus on behavioral problems regarding endocrine profiles or pathohistological evaluation revealed no significant association between specific behavior and elevated serum hormones [41] or absent luteal tissue [17]. Moreover, behavioral problems were associated with pathohistologically detected GCTs in 45% of examined mares [18]. However, the focus of these studies was set on the diagnosis of GCTs without a histomorphological and immunohistochemical evaluation of clinically unremarkable ovaries. Dolin et al. (2023) [21] reported a detailed immunohistochemical analysis in a single GCT and emphasized the need for a complete clinical data set for proper analysis. Our study involved a clinical evaluation of all mares, including the initially presenting problem, conservative treatments, clinical examination and analysis of serum AMH and testosterone concentrations before surgery, surgical outcome, and an owner´s survey of each case, with a following pathohistological evaluation of the removed ovaries. We could determine ENCs and/or anovulatory-like follicles in 35% of bOE as ovarian abnormalities and suggest them as a pathohistological explanation for the evaluated postoperatively high success rate in bOE. The remaining 65% of mares in bOE revealed no pathohistological findings, but 85% of them (11/13) improved after BO. We therefore conclude an individual hormonal influence, which might not be evaluable [41], or a detection failure of pathological ovarian changes by routine diagnosis, as demonstrated in the three cases with ENCs. The detection of microscopically small ENCs by pathohistology could, therefore, be missed in other cases, which might result in an even higher incidence of these ovarian abnormalities in mares with behavioral problems. This fact would strengthen the justification for BO due to medical necessity [11]. However, BO can also be regarded as medically necessary in the case of severe clinical problems like colic symptoms. In the case of aggressive behavior, animal welfare reasons (i.e., housing) justify a BO. In conclusion, further investigations into ENCs, their impact on serum hormone concentrations, and their association with behavioral problems are warranted. Moreover, improvements of routine diagnostic methods are necessary to detect early ovarian abnormalities and further decide which ovary to remove, especially if breeding is not excluded. Diagnostic accuracy might be achieved by means of other diagnostic markers like CD56, GATA-4, or FOXL2, which are indicative of the prognosis of human GCTs [40].

## 5. Conclusions

Bilateral ovariectomy resulted in a high owner satisfaction in 85% of mares with clinically unremarkable ovaries but behavioral problems and therefore confirmed previous reports. Serum AMH and testosterone were indicative of advanced GCTs but variable in mares of bOE. Immunohistochemical analysis of Ki67, AMH, AR, EGFR, CAL, and E-Cad was not helpful in differentiating between clinically unremarkable and GCT-affected ovaries and needs further investigations. However, pathohistological evaluation of bilaterally removed ovaries revealed clinically nondetectable ENCs in 15% and anovulatory-like follicles in 30% of mares with behavioral problems in bOE, which were regarded as precursor stages of neoplasia. These structures might be a pathohistological explanation for behavioral problems of ovarian origin in mares and a reason why BO may result in behavioral improvement. 

## Figures and Tables

**Figure 1 animals-14-02899-f001:**
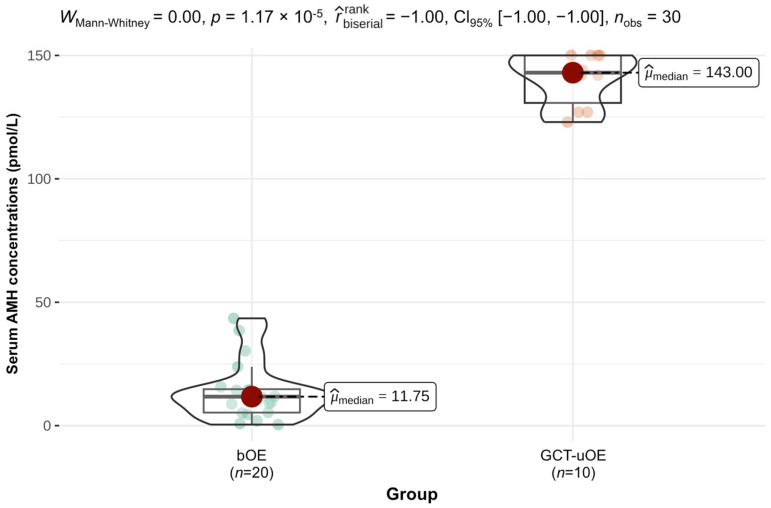
Comparison of serum anti-Müllerian hormone (AMH) concentrations between bilaterally ovariectomized mares (bOE, *n* = 20) and mares with granulosa cell tumors (GCT-uOE, *n* = 10). Mann–Whitney U (Wilcoxon rank-sum) nonparametric test was used due to not-normally distributed data (Shapiro–Wilk normality test). *p*-value < 0.001 shows very strong evidence for the (rank-sum) difference in serum AMH concentrations between bOE and GCT-uOE. Rank biserial correlation coefficient of −1.00 with 95% confidence interval indicates a very large effect size of serum AMH concentrations between the two groups. The red bullets show the median AMH concentrations; the green bullets show AMH concentrations of single cases. *n*_obs_ = total number of tested mares; *x*-axis: analyzed different groups, bOE = mares with bilaterally removed, clinically unremarkable ovaries, *n* = 20; GCT-uOE = mares with unilaterally removed granulosa cell tumors, *n* = 10; *y*-axis: measured serum AMH concentrations in pmol/L of each case with a range of 0.4–150 pmol/L.

**Figure 2 animals-14-02899-f002:**
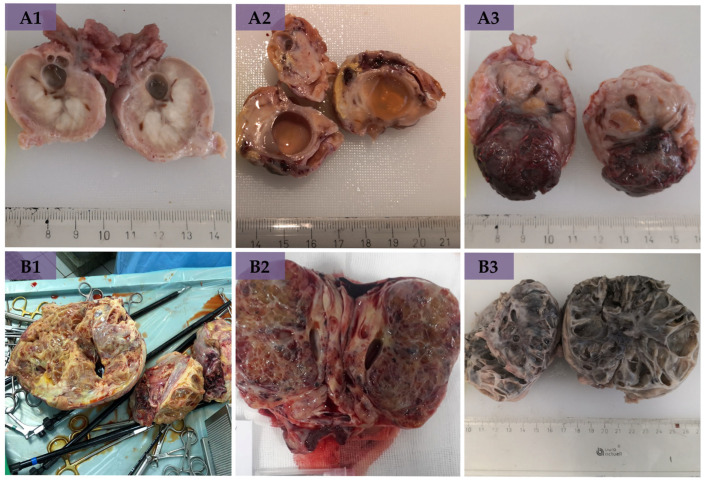
Ovarian gross sections of bilaterally ovariectomized mares (bOE, (**A**)) and mares with granulosa cell tumors (GCT-uOE, (**B**)). (**A1**,**A2**) Ovary after fixation with preantral and tertiary follicles. (**A3**) Ovary with a corpus luteum (CL). (**B1**) Fragmented 5 kg granulosa cell tumor (GCT) immediately after removal via laparotomy. (**B2**,**B3**) GCT with typical multicystic appearance.

**Figure 3 animals-14-02899-f003:**
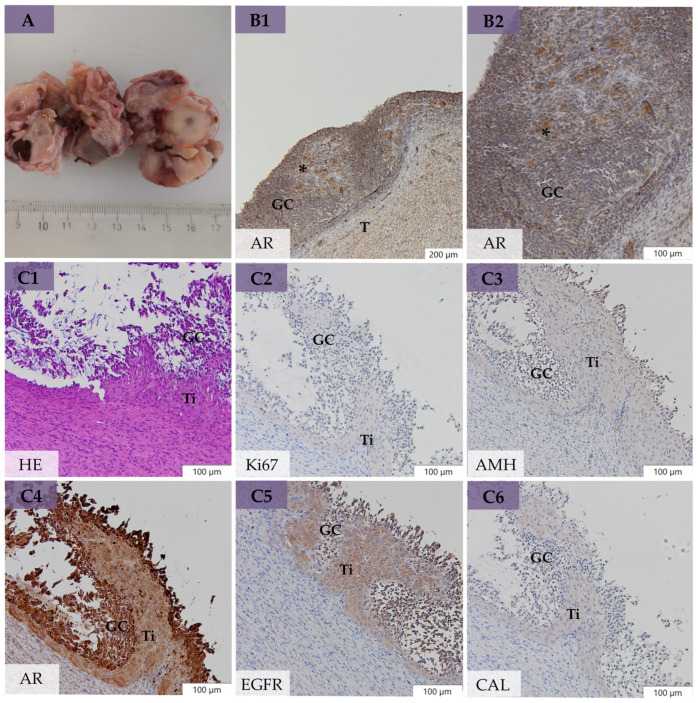
Early neoplastic changes (ENCs) in different ovaries of bilaterally ovariectomized mares (bOE, Case 29 and Case 10): (**A**) Gross section of a clinically unremarkable ovary with a pale area (red circle) near the ovulation fossa suspicious for ENCs (Case 29). (**B**) Immunohistochemical evaluation of this area in aromatase (AR) staining in different magnifications (**B1**,**B2**); note the granulosa cell (GC) nests with AR-positive cells resembling Leydig-like cells (LLCs) in between (asterisk), defined as ENC; Bars 200 µm, 100 µm. (**C**) Pathohistological findings in a clinically unremarkable ovary with detected ENCs (Case 10): spindle-shaped, neoplastic GCs with polyhedral, foamy cells in the theca interna cell layer resembling LLC. Figures are presented in hematoxylin and eosin (HE, (**C1**)) and different immunohistochemical staining with Ki-67 (Ki67, (**C2**)), anti-Müllerian hormone (AMH, (**C3**)), AR (**C4**), epidermal growth factor receptor (EGFR, (**C5**)), and calretinin (CAL, (**C6**)); bars 100 µm; GCs = granulosa cells, T = theca cell layer, Ti = theca interna cells.

**Figure 4 animals-14-02899-f004:**
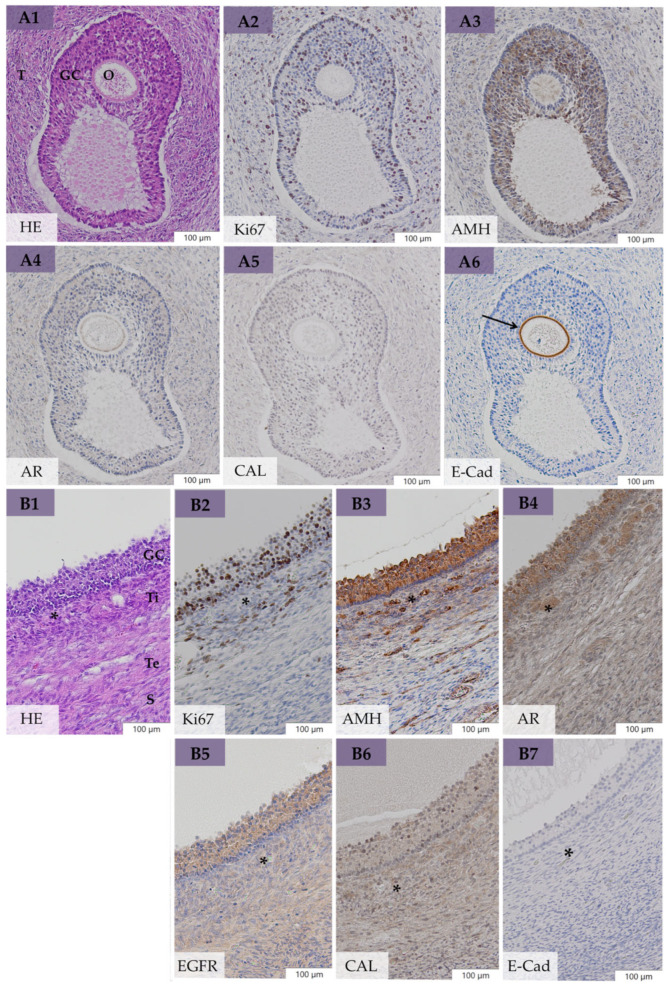
Antral follicles in ovaries of bilaterally ovariectomized mares (bOE) in different immunohistochemical staining with hematoxylin (HE (**A1**,**B1**)), Ki-67 (Ki67 (**A2**,**B2**)), anti-Müllerian hormone (AMH (**A3**,**B3**)), aromatase (AR (**A4**,**B4**)), epidermal growth factor receptor (EGFR (**B5**)), calretinin (CAL (**A5**,**B6**)), and epithelial cadherin (E-Cad (**A6**,**B7**)): (**A**) Tertiary follicle with oocyte, granulosa cell (GC) layer, and theca cell layer (T); note the positive E-Cad staining of the zona pellucida (**A6**, arrow). (**B**) Preovulatory follicle with GC, theca interna (Ti), and theca externa cell layer (Te) and stroma (S); note the polyhedral cells with foamy cytoplasm in the theca interna cell layer (asterisks); bars 100 µm; GCs = granulosa cells, O = oocyte, S = stroma.

**Figure 5 animals-14-02899-f005:**
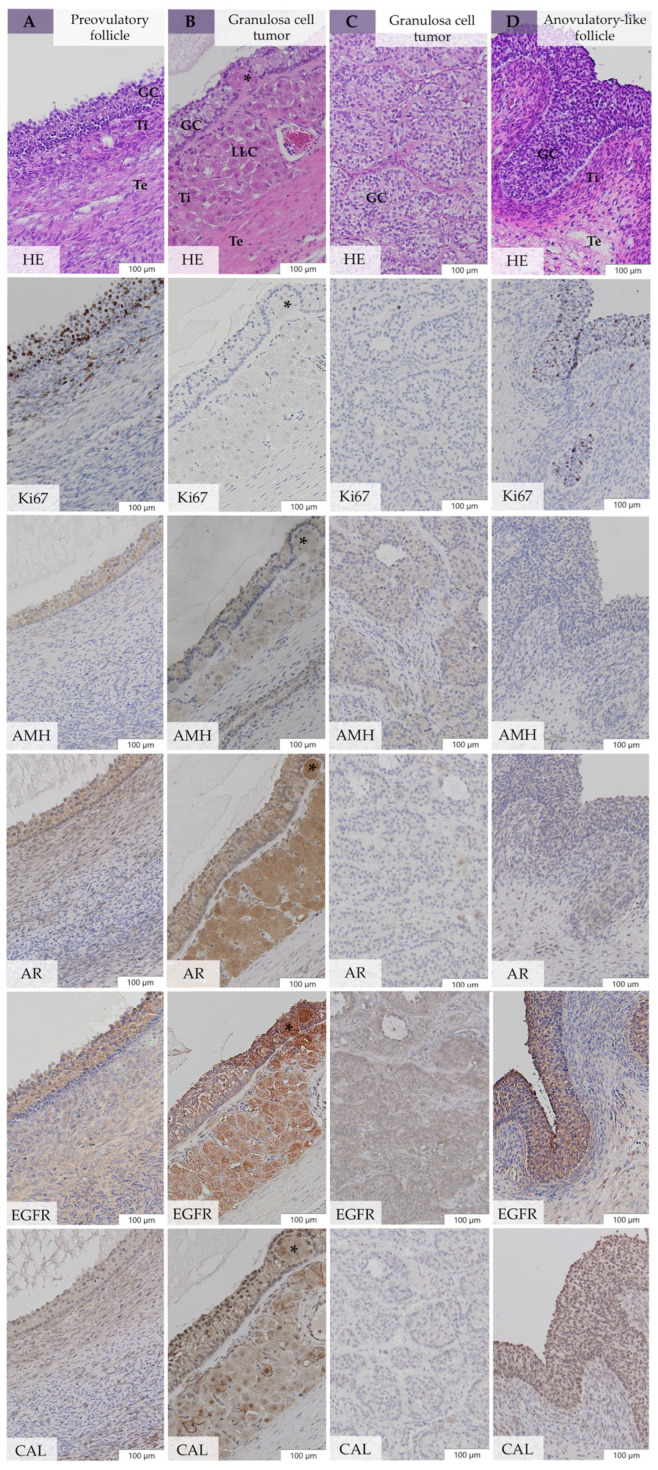
Large follicular structures in ovaries of bilaterally ovariectomized mares (bOE; (**A**,**D**)) compared with cyst-like structures in two different granulosa cell tumors (GCTs) of unilaterally ovariectomized mares (GCT-uOE; (**B**,**C**)) in hematoxylin and eosin (HE) and different immunohistochemical staining with Ki-67 (Ki67), anti-Müllerian hormone (AMH), aromatase (AR), epidermal growth factor receptor (EGFR), and calretinin (CAL). (**A**) Preovulatory follicle with granulosa cell (GC) layer, theca interna, and theca externa cell layer. (**B**) GCT with a multiple GC layer and numerous Leydig-like cells in the theca interna cell layer, some show an invasive growing character (asterisk). (**C**) GCT with microfollicular pattern of the GC layer. (**D**) Anovulatory-like follicle with a multiple GC layer, poorly developed theca interna, and theca externa cell layer; bars 100 µm; GCs = granulosa cells, Ti = theca interna cell layer, Te = theca externa cell layer, LLCs = Leydig-like cells.

**Table 1 animals-14-02899-t001:** Function and diagnostic relevance, clone, sources, dilution, pretreatments, and positive controls of used immunohistochemical markers: Ki-67 (Ki67), anti-Müllerian hormone (AMH), aromatase (AR), epidermal growth factor receptor (EGFR), calretinin (CAL), and epithelial cadherin (E-Cad).

Antibody	Function—Diagnostic Relevance	Clone/Host	Source	Dilution	Pretreatment	Positive Control
Ki67	Cell proliferation—Tumor marker	MIB-1 Mouse	Agilent Technologies, Santa Clara, CA, USA	1:1000 in PBS	0.01 M Citrate buffer pH 6.0	Lymph node
AMH	Hormone—Produced in granulosa cells	Polyclonal Rabbit	Genetex, Irvine, CA, USA	1:5000 in PBS	Tris EDTA pH 9.0	Fetal gonads
AR	Enzyme complex— Conversion of androgens into estrogens	Polyclonal Rabbit	BioVision, Waltham, MA, USA	1:1000 in PBS	Tris EDTA pH 9.0	Testes
EGFR	Transmembrane protein with function in ovulation—Tumor marker	4575 Mouse	NeoBiotechnologies, Aachen, Germany	1:200 in PBS	Tris EDTA pH 9.0	Skin
CAL	Calcium-binding protein—Tumor marker	Polyclonal Rabbit	Chemicon, Temecula, CA, USA	1:5000 in PBS	0.01 M Citrate buffer pH 6.0	Cerebrum
E-Cad	Adhering cell junction—Tumor suppression activity, Tumor marker	Polyclonal Rabbit	Sigma Prestige, St. Louis, MI, USA	1:500 in PBS	0.01 M Citrate buffer pH 6.0	Kidney

**Table 2 animals-14-02899-t002:** Frequency of behavioral problems in mares with bilaterally removed, clinically unremarkable ovaries (bOE; *n* = 20) and mares with unilaterally removed granulosa cell tumors (GCT-uOE; *n* = 10) as referred by the owner. Mares showed in general more than one specific behavioral problem. Values are expressed as percentages and ratios (in parentheses).

Behavioral Problem	Frequency of Occurrence	
	bOE	GCT-uOE
Moodiness	50% (10/20)	40% (4/10)
Stressed manner	30% (6/20)	20% (2/10)
Unwillingness to be ridden	45% (9/20)	30% (3/10)
Aggressiveness towards people	15% (3/20)	0% (0/10)
Aggressiveness towards other horses	30% (6/20)	0% (0/10)
Stallion-like behavior	5% (1/20)	60% (6/20)
Increased flank sensitivity	25% (5/20)	0% (0/20)
Colic symptoms	40% (8/20)	10% (1/10)
Prolonged or constant estrous signs	20% (4/20)	10% (1/10)
No estrous signs	5% (1/20)	40% (4/10)
No abnormal behavior	5% (1/20)	10% (1/10)

## Data Availability

The data that support the findings of this study are available from the corresponding author upon reasonable request.

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
