# Peer review of "Pathohistological Findings after Bilateral Ovariectomy in Mares with Behavioral Problems"

_animals, 2024, doi:10.3390/ani14192899_

Round 1

Reviewer 1 Report

Comments and Suggestions for Authors

Dear Dr Wolf and colleagues,

Than you for submitting this interesting manuscript for consideration for publication in Animals. This is a generally well considered and well written manuscript. 

General comments:

The objective of your study was to pathohistologically evaluate bilateral removed, clinically unremarkable ovaries of mares with behavioral problems (bOE, n=20) and to compare them with pathohistologically confirmed, unilateral removed granulosa cell tumors of mares with neoplastic ovaries (uOE, n=10). You have also used the data gained from this to justify the use of bilateral ovariectomy in the mare as a long-term solution for mares with behavioral problems despite clinically unremarkable ovaries. Although I do not disagree that there are advantages of the bilateral ovariectomy for this purpose you have not discussed the limitations of this study in relation to this conclusion.

For this conclusion to be supported you would ideally compare your pathohistological findings with behaviourally normal disease-free mares and not those with disease; in this case granulosa cell tumours); an appropriate group for comparison would have been a group of age (and parity) matched mares without behavioural reported abnormalities, they would also (in both groups) not have received any GnRH vaccines. The readers should be informed of the limitations of the study so that they can make a judgement on the validity or the strength of such conclusions. Perhaps you need to modify your objectives as the paper tries to both pathohistological comparison, and behavioural change.

In your supplementary table (S1) the mares’ ages are not included which I believe is of particular interest in the cases where early neoplastic changes were detected, nor do we have a time of year which the cases presented which is of importance when discussing the stage of cyclicity, size of ovary or even the presence of anovulatory structures. It is up to you if you want to add ages and time of year to your supplementary materials.

The document you refer to [11] to inform us that many options to treat horses medically and/or conservatively is extremely intersecting however as far as I understand it is discussing oestrus suppression in light of German animal welfare and pharmaceutical law. If my translation is appropriate it states:

“What is also significant from a liability perspective is the fact that most of the practices described below are not medically necessary, but are generally “requested treatments” by the animal owner/rider in order to achieve an increase in performance or consistent performance. These measures do not meet the veterinary standard of care.”

Please can you address why bilateral ovariectomy is not viewed in the same way as you are in many instances, less the 15% of ovaries with what you have described as having early neoplastic change (ENC), have pathohistologically normal ovaries? Is this surgery, not medically necessary? 

Specific comments:

Materials and methods:

Line 49: You state ‘…often related to estrus…’ I would suggest this is assumed and request you amend to: … assumed to be related to estrus, an association to ovarian hormonal secretion and anatomic changes is also assumed,…’

Line 98: extent not extend

line 136: allyltrenbolone not Allyltrenbolon

Lines 135-140: How were behavioural problems categorised? I appreciate you have referred to [17] & [18] however, Devick et al. (2020) used a preoperative questionnaire to gather description and duration of any specific unwanted behaviours and they listed the categories in their results (for reference it is not clear how those authors collected or categorised the behaviours in their study). Straticò et al. (2023) retrospective interviews and a telephone questionnaire were performed. Participants were asked to list the type of misbehavior and to score the misbehavior before and after surgical treatment. (A list of misbehaviours are provided but it is not clear if the participants volunteered the abnormal behaviours or they were provided with them). This causes me concern as your ‘behaviour problem’ categories are vague and could carry multiple meanings and interpretations, such as: moodiness, stressed manner, aggressiveness and stallion-like behaviour (with regard to the latter please see comment regarding lines 522-523). So how did you 1. collect information on abnormal behaviours and then categorise them into the categories you provided? Please add to materials and methods and consider adding any such questionnaire(s) to the supplementary materials.

Lines 143-146: ‘Anestrus was defined if large follicles >30 mm and CL were absent, estrus if antral follicles >30 mm were present in absence of a CL and luteal phase if at least one CL was present [41,42].’ What are references 41 & 42 referencing here? I see no reason for [41], for [42] the following is in the materials and methods: ‘The stage of the cycle was determined by examination of the ovaries. The mare was regarded to be in diestrus or estrus when a corpus luteum or a follicle ≥3.0 cm in diameter without a corpus luteum on either ovary was present, respectively.’ This is however, only in part what your categorisation was. You had mares in the ‘follicular phase’ and the ‘luteal phase’ however readers may be confused by the term anoestrus; this could be interpreted as winter anoestrus as well as the stage between the end of the luteal phase and the detected 30mm follicle. Luteal tissue can be of variable size and in reality only progesterone assay can assess the presence or absence of luteal tissue. It would be far more appropriate in the absence of progesterone to state the presence or absence of a CL and the diameter of the largest follicle or range of follicle sizes. The time of year (month) of surgery would be suggestive of the degree of cyclic activity anticipated. Please address with a reviewed description, you need not change the categorisation just the clarity of the written description.

Lines 152-155: To paraphrase, AMH assay was by means of a standardised and validated ELISA. Please can you state the assay used (manufacturer) and the methodology should be described or reference made to a published methodology. This is important as your results differ (lines 531-532), and you also state that the reference range has been adjusted (lines 549-552). The same detail  should apply to testosterone please.

Lines 178-180: An owner’s questionnaire was used but I do not see this presented in the supplementary materials; please defer to editorial view if this is required as per journal standards.

Line 186 & 189: What constituted a ‘representative sample’?

Lines 202-203: ‘Macroscopical and pathohistological evaluation of removed ovaries was performed by the same observer.’ Was this observer appropriately trained and/or experienced?

Results:

Line 283-284: ‘In bOE, mares were 4-20 years old at presentation (mean 13.0 years, SD 3.7)’ This is a very interesting parameter and coupled with (lines 303-304) ‘The time from first presentation of behavioral problems to referral reached from 1 month to 8 years with 62% under 12 months and 38% over 12 months in bOE’. If your manuscript is to continue an attempt to justify the validity or appropriateness of the surgical bilateral ovariectomy for behavioural problems one needs to consider and discuss this data. The duration of ownership needs to be considered as well as the duration of the behavioural abnormality, ie. is the behaviour new and therefore gives weight to the argument that pathology has developed (in the ovaries or other body system (musculoskeletal)) or ,if the behaviour is longstanding (in an older mare) or if the mare is young, it is therefore more likely to be related to the inherent character of the individual mare or a longstanding and perhaps pre-exiting problem. The trainer/rider/owner history with the mare is extremely important as this reviewer has seen mares behave well with one owner/rider and then behave badly with a different owner/rider.

Line 355: It is notable that 15% of cases underwent bilateral ovariectomy suggesting that the ovaries were not the cause of the problem. This should be considered in light of the conclusions drawn.

Discussion:

Lines 505-515: The discussion of the cases included that had received GnRH vaccine is lacking in my opinion and one should consider the effect of GnRH vaccine on ovarian size and activity, I appreciate that all vaccinated cases (according to S1) had no significant histomorphological abnormality (discounting fossa cysts) however is it possible that GnRH vaccination suppressed some significant abnormality such that it remained undetected?

Line 513: ‘In our study, 73% of the conservatively treated mares resulted in a positive outcome…’ This was only a small number of mares and so the percentage should ideally be accompanied with the proportion (8/11). Please include.

Lines 522-523: ‘Aggressive behavior was not observed in any case of uOE, although reported as pathognomonic for mares with GCT [49].’ One needs to understand that testosterone-driven-sexual-behaviour and stallion like behaviour can be considered synonymously by some and this is a matter of interpretation. Testosterone driven behaviour could certainly be considered as pathognomonic for GCT so it becomes a matter of perspective, what is stallion-like for one could be aggressive to another. Please reconsider this statement in light of this.

Lines 528-531: ‘However, inclusion of mares in all cyclic stages and with abnormal cyclicity (15%) or anovulatory follicles (30%) and ultrasonographically not detectable, small nests of neoplastic GC (ENC, 15%) on one or both ovaries of bOE might explain the deviating median serum AMH in our study compared to others [34,35].’ What abnormal cyclicity (15%) are you referring to? As stated previously it would have been good to see the ages of the mares that had ENC.

Line 609: safe not save

Line 637: ‘…and possible involvement in GCT development [53].’ Also [60].

Lines 651-653: ‘Similar presence of LLC and SLC in uOE with 70% were also found in other studies [22,25,49,51,55,69].’  Not [69] here.

Line 666, 681 & 710: ‘Dolin et al. (2023)…’ add [21].

Lines 735-736: ‘These structures might be one explanation for the high success rate regarding behavioral improvement after BO in bOE.’ Consider rephrasing to: These structures might be a pathological explanation for behavioural problems of ovarian origin in mares and a reason why bOW may result in behavioral improvement.

Lines 738-741: ‘However, due to the high success rate of behavioral improvement and the low complication rate, BO is regarded as efficient long-term solution for mares with behavioral problems despite clinically unremarkable ovaries.’ Please consider the weaknesses of your study design, the reliability of the data collected and the fact that it was considered a failure in 15% of mares when making this statement. I would prefer it if you concentrated on what you have determined from this study rather than trying to consider both pathohistology and behavioural change.

Comments on the Quality of English Language

The quality of the english is generally very good

Author Response

Response to Reviewer 1:

please see the attachement

Reviewer 2 Report

Comments and Suggestions for Authors

Simple summary:

line 14: write ” … compared with pathohistologically confirmed granulosa cell tumors.” Unnecessary to write that they were removed unilaterally.

Line 19: write “cyst-like”

Line 22: “Ultrasonographically non-detectable early neoplastic changes could be determined in 15% of what? Mares? Ovaries?

Line 23: “with bilaterally removed ovaries”

Line 22: What is your definition of an anovulatory follicle? Follicles that are not selected for ovulation will grow, undergo atresia and regress. Would you call these anovulatory? Or is it the follicles that will grow to a large size but do not receive the signal to ovulate from LH, as is seen during Autumn or in mid diestrus?  These follicles will disappear with time. Use another term for the follicle that you describe, if you think it is an early tumour.

Abstract:

Line 31: “and a justification for BO controversially discussed”. Rephrase. The meaning is not clear.

Line 32: “…bilaterally…”. I think it is an adverb. Goes for the remainder of the manuscript.

Line 40: write non-detectable instead of “not detectable”. Goes for the remainder of the manuscript.

Introduction:

Line 80: Remove “controversially”. Or write that it is controversial, but do not write “controversially discussed”.

Line 111: … GCTs … (plural)

112-114: The groups “bilateral ovariectomy” versus “unilateral ovariectomy” should be rephrased so the unilateral ovariectomy emphasizes that it is a granulosa cell tumour and not only a unilaterally removed ovary. Maybe just use the abbreviation GCT-OE.

Materials and Methods:

Line 126: “Minimal-invasive” should be “minimally invasive”

Lines 135-137: “A detailed history was taken prior to surgery. The information included age, breed …. and whether the mare had received altrenogest or a GnRH vaccine”. (As the sentence is now, it reads as if all mares received altrenogest and a GnRH vaccine prior to ovariectomy).

Lines 139-140: “…increased sensitivity of the flank…”

Line 140 “unwillingness to be ridden…”

Line 143: …determined by the presence of follicles and ……”

Lines 143-148: I would question the term “anestrus” when follicles were less than 30 mm and there was no CL. Anestrus to me is e.g. winter anestrus, where there are follicles less than 16 mm and GnRH, LH and FSH is low, or anestrus during the breeding season, but with small follicles. During transition you will also see follicles in their twenties and no CL. Are these mares anovulatory? In that case maybe use that term for a mare that does not cycle normally, has follicles that just do not ovulate.

Line 150: Change “levels” to “concentrations”. Also change in the rest of the manuscript.

Lines 155-158: AMH cut off limit for GCT is rather low.  Ball et al (2013) found that a cut off value of >4 ng/ml (28.6 pmol/l) is supportive of a diagnosis of granulosa cell tumour. 14.3 pmol/l (2 ng/ml) is within the normal range of AMH in Ball’s paper. Please elaborate on your choice of cut-off value.

Line 160: “From” should be “by”.

Line 161: Describe the “special diet” more. Were the mares starved?

Table 1: Which part of the fetus was used as the control?

Lines 197-198: Briefly describe how you define early and late atretic follicles.

Results:

Table S1: “Owners satisfaction after bilateral ovariectomy”. Remove “bilateral”.

Line 282: PRE: write Pure Raza Española

Line 362: “Tradable”. Use another word. It does not make sense here.

Line 371: Figure 4A, not 5A.

Line 384: How can you tell that the follicle is anovulatory? Only time will tell whether it would ovulate. It is not an atretic follicle as the granulosa cell layer is present, and hence it could have kept growing if left in the mare. Please elaborate or change the wording for these follicles.

Discussion:

Lines 629-639: Take previous comments about anovulatory follicles into account.

File with original images: Translate from German to English.

Comments on the Quality of English Language

A native English-speaker should review the manuscript for correction of language mistakes.

Author Response

Response to Reviewer 2:

Reviewer 3 Report

Comments and Suggestions for Authors

This study described possible pathology and related histological findings underlying an important issue for both the horses’ owners and veterinarians, which are the behavioral problems in mares, often related to estrus and/or ovarian pathology. As described by the Authors, besides conventional treatment, ovariectomy, especially in the case of GCT might be an option. In general the article is well written and I recommend it for publication.

Nevertheless, there are issues which, at least some of them, the Authors themselves considered limitations, and should be addressed if any further study in this area will be conducted. This include the detailed examination of the mare considering also other putative causes of unwanted behavior, especially in the case of colic symptoms or detailed objective, behavioral questionnaire. In addition, measurement of other hormones as P4 and E2 throughout the cycle, or expression of other factors might be verified.

Author Response

Response to Reviewer 3:

Round 2

Reviewer 1 Report

Comments and Suggestions for Authors

Dear Drs Wolf and colleagues,

Thank you for addressing my concerns and comments. I feel that these have been adequately addressed and have no further comments.

Yours sincerely